# Demand Coupling Drives Neurodegeneration: A Model of Age-Related Cognitive Decline and Dementia

**DOI:** 10.3390/cells11182789

**Published:** 2022-09-07

**Authors:** Josh Turknett, Thomas R. Wood

**Affiliations:** 1Brainjo Center for Neurology and Cognitive Enhancement, Atlanta, GA 30076, USA; 2Department of Pediatrics, University of Washington, Seattle, WA 98195, USA; 3Institute for Human and Machine Cognition, Pensacola, FL 32502, USA

**Keywords:** Alzheimer’s, dementia, cognitive decline, cognitive demand

## Abstract

The societal burden of Alzheimer’s Disease (AD) and other major forms of dementia continues to grow, and multiple pharmacological agents directed towards modifying the pathological “hallmarks” of AD have yielded disappointing results. Though efforts continue towards broadening and deepening our knowledge and understanding of the mechanistic and neuropathological underpinnings of AD, our previous failures motivate a re-examination of how we conceptualize AD pathology and progression. In addition to not yielding effective treatments, the phenotypically heterogeneous biological processes that have been the primary area of focus to date have not been adequately shown to be necessary or sufficient to explain the risk and progression of AD. On the other hand, a growing body of evidence indicates that lifestyle and environment represent the ultimate level of causation for AD and age-related cognitive decline. Specifically, the decline in cognitive demands over the lifespan plays a central role in driving the structural and functional deteriorations of the brain. In the absence of adequate cognitive stimulus, physiological demand–function coupling leads to downregulation of growth, repair, and homeostatic processes, resulting in deteriorating brain tissue health, function, and capacity. In this setting, the heterogeneity of associated neuropathological tissue hallmarks then occurs as a consequence of an individual’s genetic and environmental background and are best considered downstream markers of the disease process rather than specific targets for direct intervention. In this manuscript we outline the evidence for a demand-driven model of age-related cognitive decline and dementia and why it mandates a holistic approach to dementia treatment and prevention that incorporates the primary upstream role of cognitive demand.

## 1. Introduction—Framing the Approach

Despite an extraordinary investment of resources over the past several decades, the societal burden of Alzheimer’s Disease (AD) and the other major forms of dementia continues to grow [1]. Multiple pharmacological agents directed towards modifying the pathological alterations of AD have yielded disappointing results, and no meaningful progress in therapeutics has been made since the approval of the first cholinesterase inhibitor in 1993. Furthermore, existing pharmaceuticals at best offer modest symptomatic benefits but do not alter the course of the disease. Multiple agents that target the pathological signatures of AD, particularly the aggregation of amyloid plaques, have failed to yield meaningful clinical outcomes [2,3].

Similarly, in addition to failing to produce effective targets for therapeutic intervention, significant advances in the detail of our pathological descriptions at the cellular and molecular levels have not meaningfully enhanced our understanding of the pathogenesis of these entities. Presently, we have an overwhelming collection of facts about these conditions but lack a coherent explanatory framework for understanding their specific role in the disease process [4,5]. As such, over the years the field has pulled back from the idea that the amyloid cascade is the primary driver of late-onset AD, and either focused on alternative mechanistic pathways including tau, microglia, and additional genetic factors (such as apolipoprotein E, ApoE), or proposed alternative “dual pathway” or “probabilistic” models that focus less on pathological protein aggregation and more on common upstream drivers of disease [4,5,6,7,8].

While future planned treatment modalities may still hold promise [9], as efforts continue towards broadening and deepening our knowledge and understanding of the mechanistic and neuropathological underpinnings of AD, we believe it is imperative to re-examine the level of organization at which we consider the biological underpinnings of AD. In addition to not yielding effective treatments, none of the various biological processes that have been the primary area of focus of our investigational and explanatory efforts have been adequately shown to be either necessary or sufficient to fully explain the risk and progression of AD [5,6]. On the other hand, a growing body of evidence indicates that lifestyle and environment represent the ultimate level of causation for AD and age-related cognitive decline (ARCD) and that, specifically, the decline in cognitive demands over the lifespan plays a central role in driving the structural and functional deteriorations of these entities [10,11,12,13,14,15]. 

In an attempt to provide a comprehensive theoretical model at the appropriate level of biological organization, we propose the demand-driven model of age-related cognitive decline and dementia. In the absence of adequate cognitive demand, physiological demand–function coupling leads to the downregulation of growth, repair, and homeostatic processes, ultimately resulting in deteriorating brain tissue health, function, and capacity. In this setting, the heterogeneity of associated neuropathological tissue hallmarks then occurs because of an individual’s genetic and environmental background and are therefore best considered downstream markers of the disease process rather than specific targets for direct intervention. In the following text, we initially examine the current inadequacies of the current model of AD pathogenesis and the compelling need to re-focus our investigational and explanatory efforts on the driving role of lifestyle [10,16]. We then describe the human and preclinical data that support the demand model, including the physiological processes that occur in the absence of adequate demand that result in decreased capacity and function. We describe the intrinsic and extrinsic forces for dementia risk as well as the broad clinical observations explained by this model. Finally, we address some potential methods for intervention and critical future directions for the field considering the proposed model.

## 2. Inadequacy of Current Model of Alzheimer’s Disease

It has become increasingly clear that there is considerable heterogeneity in the clinical entities that comprise AD as it is currently conceived. The differences in the clinical course, neuropathological, neuropsychological, and neuroimaging features between early and late onset forms of AD are so substantial that it is now commonplace to treat them as distinct clinical entities [17]. Since a critical component of defining a disease process is ensuring the application of accurate and appropriate terminology, we propose a distinct differentiation between classical early-onset or familial AD and “late-onset” AD.

We define late-onset AD as age-related dementia (ARD) and the period of cognitive decline that invariably precedes it as age-related cognitive decline (ARCD). When initially described, AD was a progressive dementing process that was clinically detectable when the individual was in their 3rd–5th decades of life. Indeed, the first case described by Alois Alzheimer, Auguste Deter, was supposedly found to have a mutation in the presenilin (PSEN) 1 gene [18], though this has also been disputed [19]. Classical familial/monogenic AD has since been determined to be caused mainly by single genetic mutations in three genes—PSEN1, PSEN2, and amyloid precursor protein (APP). Although the pathological “hallmarks” of classical AD and ARD share significant overlap—including aggregation of amyloid plaques and hyperphosphorylated Tau tangles—this does not necessarily mean that the upstream processes are similar enough for these diseases to fall under the same terminological umbrella (Table 1). Similarly, once they reach their late stages, early- and late-onset AD share multiple phenotypic commonalties, presumably due to a final convergence of cerebral atrophy and loss of functional capacity that results in similar effects on higher cognitive functions.

In the setting of ARD but not monogenic early-onset AD, we and others have previously argued that accumulation of amyloid plaques and other neuropathological “hallmarks” perhaps represent an epiphenomenon of neuronal stress rather than the critical etiological processes that are suitable as targets for direct intervention [3,20]. This suggestion is underscored by the fact that plaque burden is poorly associated with “late-onset” AD severity and progression, in both humans and animal models [21]. Though a thorough examination of the amyloid cascade hypothesis is beyond the scope of this review, it must be noted that multiple alternative models are emerging to attempt to explain the wide variety of plaque burden seen relative to an individual’s trajectory of cognitive function [5,6,22].

**Table 1 cells-11-02789-t001:** **Comparison of classical AD and ARCD based on clinical course and risk factors.** When examining the background and clinical course of classical AD and ARD, it becomes clear that they describe entirely different disease processes that should be considered separately. PSEN—presenilin. APP—Amyloid precursor protein.

	Early-Onset (Classical) Alzheimer’s Disease	Age-Related Dementia
**Age of onset**	30–60 years	>60 years
**Genetics**	Monogenic/familial: PSEN1, PSEN2, APP. ~40–50% of mutations are sporadic.	Polygenic. Most significant risk gene (ApoE) contributes ~5% of total risk and is variably penetrant based on context and environment.
**Clinical course**	Homogeneous	Heterogeneous
**Effect of environment**	Minimal, or poorly described.	Substantial. Education, diet, exercise, cardiovascular and metabolic health, history of trauma, sleep, stress, pollutants, smoking, and alcohol all have a documented role.
**Prevalence**	~1% of all AD cases and decreasing (previously estimated to represent 5% of all AD) [23].	~10% of individuals >65 and increasing (projected to double by 2050) [24]. ~99% of all AD cases [23].

Rather than continue to focus on phenotyping AD according to protein aggregates [25], an alternative approach is to acknowledge that genetic and other factors may contribute to the heterogeneity of pathological findings and instead focus on common upstream factors that coalesce around an increased risk for ARD. Abundant evidence indicates that the late-onset form of AD results from a complex mix of gene–environment interactions, with no single gene, environmental factor, or pathological hallmark demonstrated to be both necessary *and* sufficient for disease onset and progression [10,26,27,28,29,30]. Significant variations in incidence across populations, including a relative absence in indigenous and hunter-gatherer societies, also suggests that there are sets of environmental exposures that are required to express the AD phenotype even in the setting of high genetic risk [31,32,33]. In fact, the case appears quite clear that the pathways underlying familial early-onset AD and ARD must now be considered separately, with decreasing weight being placed on the amyloid cascade in ARD and a greater appreciation of other upstream factors [5]. As we will discuss, identifying the appropriate level of biological organization for a theoretical framework is essential for generating both a coherent pathogenetic model and effective therapies. Within this article we will therefore only consider a model for ARD, as true familial AD must be considered a separate entity. 

Altogether, the failure thus far to identify a suitable molecular target and the absence of a unifying pathophysiological framework despite decades of research, and the substantial evidence of a central role for lifestyle and environment in the pathogenesis of these conditions indicates that the reductive approach to generating both a pathogenetic model and therapeutic interventions is flawed. This highlights an urgent need to reassess the fundamental approach guiding our analytical, therapeutic, and explanatory efforts, for which the first step should be the identification of the appropriate level of biological organization towards which to direct them. 

## 3. Hierarchies of Explanation and Observations That the Model Must Account for

Every biological phenomenon can be described and characterized at multiple levels of biological organization, from the level of cells and molecules to that of populations and ecosystems [34]. Every biological phenomenon also has a level of analysis and characterization most appropriate for coherent explanation. This includes the level of analysis best suited to a coherent explanation of disease pathogenesis, which is typically the level most upstream in the causal chain. Characterizations of downstream consequences at lower levels of biological organization (e.g., formal neuropathology), regardless of their level of detail, do not necessarily confer additional or greater explanatory power. 

In the case of disease, the appropriate level for a coherent explanatory model will also be the optimal level of influence for effective therapeutic intervention. Targeting downstream pathophysiological alterations, in addition to being highly unlikely to yield meaningful clinical benefits, is also fraught with risk, especially in the absence of a coherent model that can clearly delineate which downstream events are pathogenic, epiphenomenal, or compensatory. The failure to identify the appropriate biological level will then perpetually frustrate explanatory and therapeutic efforts [35].

Research on the pathogenesis of ARD has focused largely on the goal of providing an explanatory model at the level of cellular and molecular mechanisms and their histopathological outputs, partly in the hope of finding a molecular target suitable for pharmaceutical intervention, and partly due to initial suspicions of a genetic etiology. As mentioned briefly above and expanded upon below, the concept of ARD as a genetically driven condition, unlike traditional AD, is no longer tenable. Moreover, the multitude of pharmacological agents investigated over the last half century have proven to be harmful, inert, or minimally helpful—an outcome that would be expected of interventions that target downstream events that do not meaningfully contribute to disease pathogenesis. 

At the same time, a considerable body of evidence has emerged indicating that ARD is environmentally driven, and that lifestyle-based interventions can substantially alter its clinical course [36]. As such, our analytical and therapeutic efforts and resources should be directed primarily towards the organismal—rather than cellular or molecular—level of biological organization, as only explanations and interventions at this level are likely to yield a coherent explanatory model or satisfactory therapies. While we cannot fully discount the possibility of future success targeting the molecular “hallmarks” of ARD, the balance of current evidence suggests that continued efforts directed at the wrong level of intervention would be expected to lead to perpetual stagnation on all fronts.

## 4. Population-Level Observations and Evolutionary Theory Derive the Demand-Driven Model of Age-Related Cognitive Decline

If we accept that the current level of focus for intervention in ARCD and subsequent ARD may not be appropriate, this allows us to zoom out to a higher level of organization and identify an upstream common factor with greater overarching biological and mechanistic support—environmental conditions that underpin the long-term development, structure, and function of the brain.

As intimated above, there is abundant evidence that ARCD and subsequent ARD are driven by lifestyle and the environment, with the environment even determining whether “fixed” risk factors are penetrant. For instance, in Western cohort studies, Apolipoprotein E (ApoE) genotype is an important risk factor, determining ~6% of overall risk of ARD [30]. Those who are ApoE 4/4 homozygous have 6–30-fold increased odds of ARD when compared to those with ApoE 3/3 genotype [29]. However, it is well known that most individuals with ARD are not ApoE4 carriers, and not all ApoE4 carriers develop ARD. This finding suggests that some other upstream factor initiates ARD, even in the setting of significant genetic risk. Indeed, although ApoE is the most well-known genetic determinant of ARD, no single genetic polymorphism is either necessary or sufficient to drive the development of ARCD. The critical moderating effect of the environment is perhaps best displayed in non-Westernized populations, some of which have shown either no effect of ApoE4 on ARD incidence, or a protective effect against ARCD in certain scenarios such as chronic parasitic infection [31,32,33].

The environmental component of ARCD and ARD is well-established in large cohort studies in multiple countries but has yet to form the basis of ARD prevention or treatment policies [10,37,38,39,40]. This includes significantly increased risk of ARCD and ARD as well as worse cognitive function associated with sedentary behavior and reduced physical activity [37,41,42,43], chronic sleep deprivation or sleep disruption [37,44,45], insulin resistance and dysglycemia [46,47], poor-quality nutrition and associated changes in nutrient status [10,48], body composition [49,50,51,52], sex hormone status during and after the menopausal transition [53], and the effect of social isolation and poor social support [54]. Early evidence is mounting to suggest that lifestyle and environmental modification has the potential to prevent or even reverse ARD, especially when initiated early in the ARCD process [10,55,56,57]. Though strategies for implementation are beyond the scope of this manuscript, moving focus away from heterogeneous cellular-level pathological processes and onto environmental factors that appear to be the most upstream antecedents of ARCD allows us to better understand the disease process as well as determine intervention points that are likely to translate into reductions in the ARCD and ARD disease burden. While our model below proposes that cognitive demand may be the critical upstream factor driving the reliance and influence of other environmental factors on cognitive outcomes, we must acknowledge the importance of overall lifestyle and the environment on the risk of ARD and ARCD [10].

Changes in diet, sleep, social structure, and physical activity are perhaps the core factors that differentiate the Westernized/modernized human environment from the environments in which human populations evolved. This environmental mismatch and absence of expected environmental and physiological input/stimuli drives physiologic deterioration and ultimately disease by compromising the bodily environment necessary to support neuronal and glial function. Here, we outline a second mechanism and its association with increased risk of ARCD and ARD involving the stimulus or inputs to which the brain responds and adapts to improve or maintain global processing.

## 5. Demand as a Critical Stimulus and Initiator of Repair

The ability to make structural and functional adaptations in the face of changing environmental demands is a signature feature of biological systems, a mechanism that helps to ensure the optimal utilization of energy via the reallocation of resources according to needs. Similarly, as we better understand the processes of biological aging, one overarching commonality across aging tissues is the loss of homeostatic and cellular repair processes [58]. Repair of damage to proteins, DNA, lipids, or other cellular components is a critical aspect of maintaining capacity, with damage inexorably accumulating over time as the body ages [58]. Multiple regulatory processes are involved in the maintenance of cellular function, with autophagy in particular receiving increased attention as a critical component to regulating organ function both in the brain and peripheral body [59]. As neurodegenerative conditions are broadly associated with the epiphenomenon of abnormal protein aggregation and organelle dysfunction, cellular autoregulatory processes including recycling and repair appear to be a critical step in the accretion of pathological cellular processes and structures. However, rather than focus on these dysregulated processes as potential intervention points, as has been suggested by others [60,61], we question whether common upstream factors can be implemented to better regulate these processes in a pleiotropic manner rather than attempt to mechanistically intervene in individual pathways which are incompletely characterized and understood. 

A common factor across growth, regulatory, and repair processes is that they are upregulated in the face of increasing demand [62]. To draw commonalities across physiologic systems, we will define demand as the application of a stimulus to a tissue, such as skeletal muscle or the brain, that requires adaptation in the form of structural or biochemical changes to increase function above current abilities. In humans, demand-driven adaptations have been observed and described extensively in the musculoskeletal and cardiorespiratory systems in response to resistance and aerobic training, respectively. With resistance training, consistent heightened demands on the musculoskeletal system leads to a range of adaptations in tissue structure and function. These include muscular hypertrophy, increased vascularization, mitochondrial growth and biogenesis, an increase in glycogen storage capacity, improvements in fatty acid oxidation, increases in myoglobin content, and increases in the size and strength of connective tissue [63]. With aerobic training, consistent heightened demands on the cardiorespiratory system also leads to a range of adaptations that substantially impact tissue structure and function. These include an increase in the size and number of mitochondria, increases in VO_2_ max and the lactate threshold, increase in stroke volume at rest and an increase in cardiac output at or near maximal rates of work, and a reduction in peripheral vascular resistance [64]. In addition to the direct stimulus for growth and adaptation in function, it is well established that demand through mechanical loading or metabolic stress are key drivers of the upregulation of autophagy, and initiation of growth and repair [65,66,67].

Similarly, animal models of neurological disease—both in neurodegeneration and acute neurological injury—demonstrate that increasing cognitive demand upregulates cellular repair and autophagy processes, resulting in neuroprotection. This effect is particularly noticeable in studies that implement exercise or environmental enrichment strategies, both of which are routinely shown to decrease neurodegeneration in the preclinical literature. Environmental enrichment for rodents generally refers to the inclusion of toys or exercise opportunities to the home cage to increase play and exploratory behavior. In models of stroke, traumatic brain injury (TBI), and multiple neurodegenerative conditions, providing environmental enrichment or an exercise intervention directly increases autophagy/reparative processes in the brain, resulting in neuroprotection [68,69,70,71,72]. Indeed, although fasting is increasingly studied as a mechanism through which autophagy can be upregulated, studies in rodents show that aerobic exercise upregulates these processes much more rapidly across multiple organs, including the brain [73,74,75,76,77]. Parallel studies show the same to be true in human skeletal muscle, with relatively short periods of exercise increasing autophagy in skeletal muscle to a similar extant as several days of fasting [78,79]. An increasing body of evidence in humans parallels that found in rodents, where cognitive stimulus can result in increased brain volumes and associated improvements in cognitive function, with capacity for neurogenesis in brain regions such as the dentate gyrus of the hippocampus still seen even in older adults [80,81,82]. Conversely, removing cognitive stimuli from the environment in rodent models accelerates or exacerbates neurodegeneration or cognitive decline, making environmental enrichment an essential factor in research animal welfare. Mechanistically, this decrease in demand may be expressed as a decrease in mitochondrial capacity for energy production, with both human and animal studies of cognitive decline showing a decrease in cytochrome C oxidase [83,84,85,86]. Decreased demand can be considered to result in a coordinated downregulation of mitochondrial function in a match to meet lower demands with an appropriately lowered energetic capacity [87,88]. Demand-driven adaptations are therefore dynamic and bi-directional (Figure 1).

Altogether, these adaptations to physical or cognitive demand result in an expansion in the environmental conditions under which homeostasis can be maintained, also known as the homeostatic capacity. Importantly, enhancements in homeostatic capacity can only be generated by increasing tissue demands. As any athlete can attest, simply increasing the supply of available substrate via diet or supplementation is insufficient to drive the necessary performance-related adaptations. Although often exploited for performance-related aims in the current era, these demand-driven adaptations, including experience-driven structural reorganization across cortical tissues, likely evolved to support the maintenance of homeostasis in the face of changing environmental conditions. For humans, this flexibility has been central to our ability to thrive in a range of ecological niches [89].

Whereas sustained increases in the demands placed on a biological system can lead to a range of structural and functional enhancements, sustained reductions in demands lead to a loss of those adaptations and enhancements [90,91]. Again, this bidirectionality is necessary for tailoring homeostatic capacity to a particular environment while maintaining the efficient allocation of resources. Yet, there are clear circumstances in which these demand-driven deteriorations are maladaptive, resulting in marked reductions in homeostatic capacity and overall function. For example, prolonged bed rest due to hospitalization leads to rapid and significant deterioration in the cardiorespiratory and musculoskeletal systems, in addition to declines in gastrointestinal, metabolic, and cognitive function [90]. Nearly half of normal strength is lost within just 3–5 weeks of immobilization, accompanied by significant muscular atrophy—respiratory muscle weakness leads to a restrictive impairment with 25–50% reduction in respiratory capacity, and bone mass and density drop significantly, predisposing to fractures of the vertebra, femur, and distal radius [91]. These and other maladaptive responses drive the significant increase in mortality risk that follows prolonged hospitalization, in many instances secondary to conditions other than the inciting illness [92]. Similarly, hip fractures in the elderly are associated with a one-year mortality risk of 20.7%, a phenomenon related almost entirely to the demand-driven deterioration and resultant drop in homeostatic capacity, rather than the initial trauma [93]. Though they may be exacerbated with aging, these changes upon removal of physical stimulus are seen regardless of age. For example, young male athletes placed on bedrest for up to 3 weeks demonstrate similar multi-systemic deteriorations, including profound decrements in cardiorespiratory function, glucose intolerance and hyperinsulinemia, negative nitrogen balance, negative calcium balance and loss of bone mass and bone mineral density, marked reduction in VO_2_ max, and significant decreases in muscle fiber size and strength [94]. As such, it can be inferred that decreased physical capacity in the face of decreased demand is a coordinated coupling response that occurs at any time in the life course of an individual, rather than being something specific to the process of aging.

Not surprisingly, given its ubiquity in biological systems as well as the high energetic costs of brain tissue, the phenomenon of demand coupling has also been shown to exist in the nervous system. The degree to which the sensory cortices can create and maintain detailed, high-level sensory representations is contingent upon the resolution of sensory data that are acquired via primary sense organs. A reduction in the resolution of incoming sensory information thus reduces the processing demands on primary cortical sensory tissues and downstream association areas. Indeed, sensory impairments are associated with an increased risk of dementia in humans, which is then reversed when those senses are restored. Visual loss from age-related macular degeneration, cataracts, and diabetes-related eye disease are each associated with an increased risk of incident dementia [95]. A recent study showed that the increased risk with cataracts is reversed after corrective surgery [95]. Hearing loss amongst those age 45–64, irrespective of etiology, is associated with a more than doubling of dementia risk [96], with parallel studies in preclinical models confirming the effects of sensory loss on accelerated pathological decline of the brain [97,98,99,100,101]. Likewise, an association between peripheral nerve impairments and dementia risk has also been established, even when corrected for age and other health problems [102,103]. As a single unifying and well-established mechanism, demand coupling is sufficient to account for these associations, and as such represents the most parsimonious explanation for them; cognitive stimulation through sensory inputs is critical for the maintenance of tissue health and cognitive function and, importantly, that the risks associated with the loss of this input are reversible.

As is the case in the musculoskeletal and cardiorespiratory systems, the structural and functional benefits generated by sustained increases in cognitive demands are also likely mediated by direct repair and restoration of tissue structure and function. Furthermore, significant and sustained reductions in cognitive challenge would be predicted to produce pathological deteriorations in tissue structure and function, accompanied by a significant loss of homeostatic capacity. It has long been known that removal of a neuronal stimulus or connection results in atrophy or death, and removal of downstream connecting neurons. The most extreme example is in Wallerian degeneration, where severing a neuronal connection results in loss of downstream neurons via activation of the smad pathway [104]. Smad proteins have since been linked to regulation of both neurogenesis and regulated apoptosis of neurons during pruning [105,106]. This process is supported by regulatory glial functions such as synaptic pruning and support by microglia. Elimination of unneeded neuronal connections is critical to normal brain development but is also activated in the setting of neurodegeneration or after acute injury [107]. As such, it seems clear that “unnecessary” neuronal connections and brain functions are eliminated, much as unnecessary muscle tissue atrophies during bed rest or after injury, and this is an active process that can be prevented by maintaining the stimulus.

The opposite is also true—in animal models of age-related cognitive decline, sustained increases in cognitive demands have been shown to reverse widespread age-associated deteriorations in tissue structure and function [108]. Multiple human studies have demonstrated that cognitively active lifestyles are associated with a lower burden of age-related neuro-pathologies [109,110,111]. Taken together, the evidence indicates that efforts to maximize cognitive demand throughout the lifespan are clearly essential to maintaining brain tissue structure and function and preventing cognitive decline.

## 6. Interventions—Stimuli with Known Benefits on Cognitive Headroom/Brain Aging

As stated, rather than representing appropriate adaptations to reduced environmental demands, the pathological deteriorations and related rise in mortality following scenarios of prolonged inactivity indicate a maladaptive loss of homeostatic capacity. As such, there appears to exist a threshold of activity beneath which tissues are no longer appropriately maintained, leading to the emergence of maladaptive deterioration. Furthermore, the existence of pathological structural and functional deteriorations in the setting of reduced physiologic stressors (bedrest, immobilization) are best explained not by an excess of injury from exogenous sources or from an inadequate supply of substrates required to support basal tissue maintenance, but rather by a pronounced downregulation of the repair and recovery mechanisms necessary to do so. In sum, it is clear that the health and function of these tissues is directly regulated by the demands placed upon them, a phenomenon we will refer to as demand coupling [13,14,15].

A growing body of evidence demonstrates that consistently engaging in cognitive demanding activities protects against cognitive decline and dementia. Explanations for this phenomenon have typically invoked the concept of cognitive “reserve,” which is usually described as a buffering or countervailing force that mitigates the impact of the primary factors causing tissue dysfunction and deterioration [112]. In this view, cognitive challenge leads to more robust and widely distributed neural representations of knowledge and concepts, reducing the functional consequences of disease-related tissue degradation. Yet, the phenomenon of demand coupling offers a separate mechanism directly linked to the individual’s long-term cognitive function. These beneficial adaptations can be stimulated by a variety of lifestyle-related activities and skills that form a natural part of human development, including formal cardiovascular or resistance exercise, coordinative movements including yoga and dance, music, and language (Table 2). The underlying mechanisms through which these interventions may prevent, or reverse, cognitive decline include direct neurological stimulus and direct increase in demand, with peripheral and central benefits that support neurological repair and function. For instance, improvements in cardiovascular fitness correlate with production of brain-derived neurotrophic factor (BDNF), which may be driven by production of lactate and ketones during exercise [113,114,115]. 

Physical activity is also associated with improved synaptic integrity and a reduction in microglial number and activation, particularly as pathological hallmarks of disease accumulate [116,117]. Similarly, increased social connectivity and support is associated with lower systemic inflammation, greater production of neuroprotective neurotransmitters such as oxytocin, and reduced activation of the hypothalamic–pituitary–adrenal and sympathetic axes [118]. A more holistic view of these interventions could therefore be that they are not disparate lifestyle factors that should be addressed or investigated in isolation, but a range of beneficial stimuli that have pleiotropic benefits and overlap in the critical aspect of cognitive demand, and which directly influence the process of neurodegeneration. 

**Table 2 cells-11-02789-t002:** **Potential interventions.** Environmental and lifestyle exposures and stimuli associated with reduced risk of age-related dementia, all of which are likely to provide a significant proportion of their benefit through cognitive demand.

Intervention/Exposure	Evidence
Exercise	Dose-dependent increase in cognitive function with minimal effective dose around 25 or 15 min of moderate or vigorous physical activity, respectively, per day [42]. Increased hippocampal volume in older adults with walking intervention [113]. Increased cognitive function in older adults after resistance training intervention [82,119]. Greater benefit from coordinative exercise such as dance, likely due to greater cognitive demand [120,121].
Language	Improved cognitive function in older adults learning a language [122,123]. Protective effect of bilingualism on the age of onset of AD [124]. Lifelong bilingualism maintains white matter integrity in older adults [125].
Music	Decreased biological brain age in musicians, with greater benefit in amateur musicians, likely due to greater demand [126]. Lifelong musicianship associated with greater white matter integrity [109]. Playing a musical instrument associated with reduced risk of dementia [127]. Older adults who played a musical instrument demonstrated higher processing speed, attention, learning, episodic memory, working memory, and executive functions [128].
Brain Training	Structured cognitive training results in improved structural and functional brain connectivity, as well as improvements in executive function and memory [82,122,129,130,131,132].
Social Connection	Increased dementia risk with lack of social support [133]. Loneliness associated with lower brain volume and up to 3-fold increase in dementia risk [134].
Sensory Input	Increased AD risk in those with cataracts, which is reversed with cataract removal [95]. Increased risk of cognitive decline with loss of sense of smell—COVID as a recent example [135,136]. Peripheral nerve impairments associated with higher risk of dementia after adjustment for age and other health factors [134]. Hearing loss associated with a doubling of dementia risk in those age 45–65, which may be reversible with hearing aids [96,137].

## 7. Cognitive Demands across the Lifespan

Due to the well-described phenomenon of demand coupling, sustained reductions in the demands placed on biological tissues alone can lead to pathological deteriorations in their structure and function. As such, sustained and significant reductions in the demands placed upon the nervous system would be expected to produce pathological deterioration. Here we review three key forces which together lead to a progressive and significant reduction in nervous system demands over the typical modern human lifespan. 

### 7.1. Genetic Forces

The human brain doesn’t reach its mature state until the third decade of life [138]. This extended period of childhood development exists in order to allow for the development and maturation of the sensory systems, along with the acquisition of multiple complex cognitive and motor capabilities that comprise the typical profile of adult cognition and behavior. The development of these capabilities requires the consistent engagement of large-scale, distributed, bi-hemispheric cortical networks throughout early life, as well as the ongoing growth and structural remodeling of their neural substrates. Furthermore, while the development of these cognitive capabilities is flexible and adaptive, the universal and stereotyped nature of their development indicates that the motivation to acquire them and the algorithms for their acquisition are genetically driven. From birth to early adulthood, these “scripted” learning sequences alone ensure a highly demanding environment for the developing nervous system. However, once these genetically driven learning sequences have completed and their corresponding cortical networks reach maturity, the acquisition of additional cognitive and motor capabilities of similar complexity only occurs voluntarily, according to individual needs or desires. Not surprisingly, it is uncommon for adults to engage in the acquisition of cognitive capabilities that rival the childhood neurodevelopmental program.

### 7.2. Neurological Forces

A distinguishing feature of the human brain is its ability to continually acquire new knowledge and skills via the alteration of its own structure and function. The progression from the novice to advanced stages of learning is facilitated by the phenomenon of automaticity, where cognitive and motor procedures once requiring conscious attention and effort can be performed while attention is directed elsewhere. Neuroanatomically, the development of automaticity is characterized by a reduction in the neural resources required to support the particular procedure [139]. This neurophysiological fact of learning ensures that the demands on the nervous system will peak at the earlier acquisition phases of learning in a particular domain, and diminish as mastery is achieved. As such, any intervention that aims to increase cognitive stimulus and prevent or reverse cognitive decline must acknowledge that, as the individual gains mastery of a skill or modality of demand, introduction of a different or ever-increasing complexity of demand is necessary in order to continually drive neurological adaptation and repair.

### 7.3. Cultural Forces

In present-day society, educational opportunities and resources are directed heavily towards early life. In high-income countries, formal education is mandatory only from early childhood to late adolescence and early adulthood. The overwhelming majority of individuals pursuing degrees at the college or graduate level are in the second and third decades of life. This education, provided decades before any later decline in cognitive function, remains a significant protective factor against ARCD and AD, either by increasing the cognitive reserve or allowing for greater ongoing cognitive stimulus through more cognitively challenging work environments [10].

In addition to the acquisition of the complex motor skills that support movement and communication that are part of the scripted neuro-development program, the acquisition of “non-scripted” complex motor skills is also heavily biased towards early life. The opportunities for instruction and participation in sports, music, and other recreational activities are all significantly greater in early life, and it is commonly held that childhood is the ideal time to acquire skills of this nature. The conventional expectation is therefore that childhood is when complex cognitive and motor abilities are to be learned so that they may be utilized to support the needs of adult life.

Further support comes from the fact that indigenous and other human populations that do not exist in a modern cognitive environment also display very low rates of ARCD and ARD, even after taking into consideration differences in lifespan—lower in certain hunter gatherer groups and longer in areas such as the Blue Zones that have a high preponderance of centenarians and supercentenarians. These populations display multiple differences from Westernized human populations, but are similar in the way that they do not conform to the “traditional” work–life structure of repeated similar daily tasks (work) followed by a period of significantly decreased demand (retirement). Although the focus on the Blue Zones often revolves around their diets, two critical aspects of lifestyle common to these areas include the strong elements of social support and the fact that individuals in those communities have specific roles throughout their lifetimes, and “never retire” [140]. Conversely, multiple epidemiological studies in modern societies suggest that the abrupt loss of regular work-related cognitive stimulus due to retirement results in a parallel increase in the risk of cognitive decline. Even when adjusting for confounders that might explain both early retirement and cognitive decline such as comorbid medical conditions, those that retire earlier appear to experience ARCD sooner [11,12,141]. This effect is also greatest in those that have higher levels of education, where at a population level, work-related cognitive demand might be greatest and the decrease in cognitive demand after retirement might be most abrupt [140]. More broadly, those that retire earlier also seem to have an increased mortality risk, suggesting that multiple physical and cognitive inputs required for normal human physical functioning and repair are now attained through the modern work environment. While a range of cognitive stimuli have been shown to support and increase cognitive function, and prevent cognitive decline in humans, retirement provides perhaps the best acute change in cognitive demand that supports the demand-driven model, with direct consequences on the brain in terms of both stimulus and repair likely at the onset of retirement in the absence of compensatory increases in cognitive demand.

### 7.4. Summary of Forces Conspiring to Hasten Our Neurological Demise

In sum, genetic, cultural, and psychological forces align to promote and facilitate the acquisition of complex cognitive abilities and motor skills in early life, while discouraging or obstructing their acquisition in adulthood. Furthermore, the neurophysiological mechanisms of learning ensure that as those abilities and skills improve, the resources required to support their execution diminish. As such, this marked disparity in novel complex learning between early life and adulthood results in a substantial, and often underappreciated, drop in cognitive demands in adulthood. In some instances, occupational responsibilities, especially those that rely less on automated behaviors, may serve to reduce the disparity and mitigate its biological consequences. The loss of this mitigating factor may also explain the increased risk of mortality, cognitive decline, and ARD that is associated with retirement. Nonetheless, the combined and concerted effect of the forces outlined above leads to the marked and progressive reduction in cognitive demands after early life that typifies the modern-day human experience but also exacerbates the likelihood of cognitive decline with age. 

## 8. Supporting Growth and Repair in Response to Cognitive Demands

While the evidence indicates that a robust and sustained demand signal is necessary for the maintenance of cortical tissue health, responding adequately to that signal also requires the biological resources and time to respond and adapt. The musculoskeletal system’s response to resistance training provides a well-known and established example of demand-driven physiology and the interdependencies of these factors. For example, while resistance training is necessary to provide the stimulus for growth and repair, those stimulus-induced adaptations also require adequate substrate (e.g., dietary protein) and the time and physiologic conditions needed for them to occur (e.g., recovery and sleep). No amount of substrate and/or time will lead to muscular growth in the absence of a demand signal, and growth and repair in response to a demand signal will be compromised if recovery and repair mechanisms are not adequately supported.

The demand-driven model proposes a similar model for the growth and repair of brain tissue, providing an organizing framework for understanding of how lifestyle factors and comorbidities contribute to disease pathogenesis. Like resistance training, robust and sustained cognitive challenge is necessary to provide the demand signal to cerebral tissues, but an effective response to that signal is contingent upon having the necessary resources and time. Diet and lifestyle factors as well as co-morbidities that undermine the brain’s ability to respond effectively will thus hasten deterioration or lead to suboptimal responses to demand. 

Broadly speaking, in this model there are two primary means by which diet and lifestyle patterns and other comorbidities undermine demand-driven adaptations and thus contribute to disease pathogenesis. The first is by compromising the supply, delivery, or availability of the nutrient substrates and other biomolecules required for growth and repair. The second is by interfering with the amount of time devoted to rest and recovery. This pathogenetic model underscores the critical need for sleep in the preservation of brain health given its central role in waste clearance, tissue repair, and plastic reorganization in response to cognitive challenge [142,143,144,145]. The critical role of sleep in facilitating demand–function responses accounts for the emerging evidence of a significant association between poor sleep and a heightened risk of cognitive decline and dementia [37,44,45,146,147,148]. 

Given these interdependencies, only a holistic approach that attends to all relevant factors can lead to effective prevention and treatment. It is important to note that such an approach does not preclude the use of therapies aimed at ameliorating downstream pathologies, and in fact their impact may be amplified considerably in the context of a holistic strategy. Though its clinical significance presently remains uncertain, if reversing amyloid deposition can meaningfully improve cognitive function, then amyloid-targeting therapies could be utilized as part of a holistic program. In fact, as previously stated, meaningful benefit from treatments of this nature may *only* occur in the context of a program that includes attention to cognitive demands and the relevant lifestyle factors. Importantly, given the necessity of a multi-factorial approach, standardized, randomized-controlled trials with single-factor interventions are not the proper vehicle for advancing therapeutic knowledge in this arena. Adequate resources should be directed towards the investigation of individualized, adaptive, and holistic preventative and therapeutic interventions akin to the FINGER trials [57], and should include efforts to provide cognitive stimulation that rivals the neuro-developmental program of early life. 

## 9. Summary of Model, Observations, and Supportive Evidence

We currently lack an explanatory framework to interpret and understand the myriad pathological alterations that have been described in ARCD and ARDs. Investigations into their cellular and molecular mechanisms have failed to yield a comprehensive model of disease pathogenesis that explains the majority of clinical observations. The failure of this line of inquiry coupled with the body of evidence reviewed above indicates that the appropriate level of explanation, and likewise therapeutic intervention, is at that of the organism as a whole—lifestyle and environment. The evidence also indicates that these conditions are not driven by any single lifestyle or environmental factor, but rather by the combined effects of many factors, the particulars of which are likely unique to each individual case (genetics and life history). Here we propose a model at the appropriate explanatory level of lifestyle and environment that we believe provides the most parsimonious explanation of ARCD and ARD, accounts for the key observations, and points the way towards new and more promising therapeutic avenues. 

In this model, the key lifestyle factor driving these conditions is cognitive demand over time, which dictates the rate of structural and functional decline in the cortical tissues that support cognition. In broad terms, this rate of decline is driven by a dynamic equilibrium between tissue damage and repair. If equilibrium is maintained, tissue health and function are preserved. If the damage exceeds repair, tissue deteriorates. In individual cases, the set of specific pathological consequences of these upstream causes will be numerous and idiomatic, based on life experience and genetic background (Figure 2). 

Environmental mismatch may disrupt either side of the equilibrium, either by causing tissue damage and dysfunction exceeding restorative capabilities (e.g., lead exposure, atmospheric particulate pollution, or modern psychosocial stressors), or by providing insufficient support to growth and repair processes (e.g., nutrient deficiencies, sleep disorders, etc.) as stated previously. Cognitive demands influence the repair side of the equilibrium by regulating the resources devoted to tissue restoration via demand coupling. 

This model accounts for the most salient observations of these clinical entities: their absence in early life, pathological heterogeneity, lack of unique environmental exposures or risk factors, wide variation in clinical onset and characteristics, relative absence in indigenous populations coupled with explosive growth in the era of industrialization, mitigation by cognitive activity, and non-linear clinical progression:

**Absence in Early Life.** As mentioned previously, the exclusive onset of ARCD and ARD in mid-to-late life, as well as the stability of brain structure and function in early life could conceivably be explained by either the presence of harmful factors in the adult environment that are absent in the childhood environment, or the presence of protective factors in the childhood environment that are absent in the adult environment. To date, no factors that fit the first criterion have been identified. However, the disproportionately sustained and heightened demands of the childhood developmental environment, via demand coupling, is a suitable candidate for the second criterion. The consistently high demands in early life and related upregulation of restorative mechanisms foster the preservation of brain tissue that counteracts deterioration driven by environmental mismatch. A significant and progressive drop in cognitive demands, as is currently typical of mid-life and beyond, would be expected to lead to progressive cognitive deterioration, including to degrees that are maladaptive and pathological.**Heterogeneity of Pathology and Clinical Course.** The heterogeneity in pathological findings and clinical course in ARDs and ARCD can also be entirely accounted for in this model. Both the accumulation of environmental contributors over time and the cognitive demands over the lifespan will significantly modify tissue structure and function. Together, these factors would produce wide variations in the clinical course, as is observed. Generally, individuals leading lifestyles of low mismatch that include expected physiological inputs (e.g., sleep, movement, social connection) and high cognitive demands throughout adulthood would be expected to display the lowest rates of deterioration. On the other end of the spectrum, those with lifestyles of high mismatch (e.g., sleep deprived, sedentary, isolated) and low cognitive demands throughout adulthood would exhibit the most rapid deterioration.**Lack of Unique Environmental Exposures or Risk Factors.** To date, no single genetic or environmental factor unique to the diseased population has been identified, indicating that the disease state is multifactorial in origin; the result of accelerated structural deterioration over time brought about by normal life sustaining functions in a range of environmental conditions. As stated, in this model, the differential rates of deterioration are largely explained by differing degrees of environmental mismatch and cognitive demands.**Absence in Indigenous Populations, Explosive Growth With Industrialization.** The relative absence of these conditions in indigenous populations and the explosive rise in prevalence during the industrial age is explained by the impact of this era on the two primary driving factors. On an evolutionary time scale, the industrial and information ages have ushered in a rapid and dramatic shift in every major facet of human life and a broad array of environmental changes that strain homeostatic capabilities. Not surprisingly, this has resulted in a rapid and dramatic increase in environmentally driven chronic diseases, including ARCD and ARDs [149]. The agricultural, industrial, and information ages have also dramatically altered the typical profile of cognitive activity across the human lifespan. The continued technological advances and globalization that characterize recent human history have led to an overriding trend of increasing specialization in the kinds of tasks humans perform in both the home and work environment. From a cognitive perspective, this has resulted in increasing reliance on a restricted set of automated cognitive capabilities to fulfill daily obligations. The net result of these recent major shifts in human life has been an unprecedented decrease in the cognitive demands placed on aging individuals at the population level. Left unaddressed, this reduction in cognitive demands over the course of adult life, alongside an increasing lifespan as medical and societal advances reduce both communicable and other non-communicable causes of disease, is likely to compound into an ever-increasing prevalence of ARCD and ARDs, as has been widely projected [150].**Mitigation by Cognitive Activity.** It is now well established that a cognitively active lifestyle is the single most protective factor associated with a reduction in risk of cognitive decline [10]. However, this association has been most commonly explained as cognitive activity compensating for deterioration brought on by the pathological condition, rather than as a primary cause of the disease state. In this model, cognitive activity is a central regulator of tissue maintenance. The nature and scope of cognitive activity must therefore be considered a primary driver of the disease state.**Nonlinear Progression.** The phenomenon of demand coupling would also be expected to produce a nonlinear, accelerating decline after cognitive function deteriorates beneath a certain threshold. As cognitive demands drop precipitously in mid-life, the downregulation of restorative mechanisms leads to greater deterioration and dysfunction, manifested by the beginnings of pathological accumulations and declining cognitive function. This process is then moderated by environmental exposures and the degree of mismatch. Once this process of deterioration and dysfunction progresses to the point of cognitive frailty, the range of potential cognitive challenges is constrained. Like physical frailty, cognitive frailty constrains the scope of cognitive activities that can be performed independently. This reduction in the scope of cognitive activities reduces cognitive demands, leading to further reduction in tissue restoration, and a positive feedback cycle is created, producing a nonlinear, accelerating decline (Figure 3).

## 10. Future Directions, Therapeutic Implications, and Conclusions

While some attention has been directed towards the influence of lifestyle and environmental factors on disease risk, the direct influence of declining cognitive demands over the lifespan on disease pathogenesis has been overlooked. There are two primary reasons why it is critical that its role be accounted for in therapeutic investigations.

First, it is presently unknown whether the maintenance of the nature and scope of cognitive demands to a degree comparable to that of early life would be enough to prevent the disease state entirely. Likewise, it is unknown whether sustained reductions in cognitive demands alone are enough to produce the disease state. Existing programs for providing brain stimulation as a means of mitigating cognitive decline do not provide a degree of cognitive challenge comparable to that of the developmental program of early life, focusing on a restricted subset of cognitive capabilities, typically bypassing large regions of cortex, including those involved with the acquisition and maintenance of complex motor skills and social behaviors. While the modest results to date offer proof of concept, it is unlikely that programs of this nature will provide a stimulus necessary to fully halt or reverse course of these conditions. In our view, candidate activities for providing sufficient stimulus to maintain tissue integrity must provide a robust and ongoing challenge to broadly distributed, multimodal cognitive networks. The activities we propose most likely to induce the necessary demand and adaptation, such as language, music, and dance, are also those that are inherently “human,” and which not by coincidence drove the evolution of the cognitive capabilities we seek to preserve (Table 2).

Second, the maintenance of demands above a certain threshold may be a necessary condition for the preservation and restoration of cerebral tissue, and as such may preclude the success of therapeutic interventions intended to support restoration and repair of diseased tissue. In other words, efforts to ameliorate AD pathology, including agents previously tested in randomized clinical trials, may only be helpful in the presence of a sufficient level of cognitive demand. It is entirely plausible that compounds previously deemed ineffective, including those targeting amyloid deposition, could yield clinically meaningful results in the setting of sufficient cognitive demands. The failure to account for cognitive demand in study design may in fact explain the surprising failure to develop clinically meaningful disease-modifying pharmaceuticals. Continued failure to account for the upstream drivers of disease pathogenesis may guarantee continued stagnation in this realm. Furthermore, the interdependencies outlined in this pathogenetic model indicate the necessity of a holistic therapeutic approach and highlight the inadequacy of single-factor trials in the evaluation of potential treatments.

Though we believe that the demand model is holistic and explains the vast majority of observations relevant to ARCD and ARD, we must acknowledge some limitations and potential hurdles to implementation. In its current form as presented, cognitive demand is difficult to quantify, and therefore the model is difficult to test. From an implementation standpoint, one might hope that better public awareness of the importance of cognitive demand could drive behavior change and increased cognitive demand across the lifespan at the individual level; however, similar approaches to physical activity have yet to be broadly successful and personalized interventions are likely to be needed [151]. Overall, it is also not yet known whether cognitive demand is both necessary and sufficient to explain ARCD, and any current attempt to prevent or reverse cognitive decline in a holistic manner should include multi-modal interventions based on available best evidence such as that from the FINGER trial [57]. This hampers the ability to dissect the relevant contributions of different lifestyle factors, including cognitive demand, on the risk of ARCD and ARD. To develop the idea of cognitive demand into an evidence-based medicine framework, one potential idea would be to incorporate it into more modern adaptive trial designs such as SMARTs (Sequential Multiple Assignment Randomized Trials), which are increasingly being implemented in the field of lifestyle and behavior change research [152,153]. An individual’s risk in multiple areas of lifestyle could assessed, and then participants randomized at multiple timepoints to sequential new interventions including personalized interventions to increase cognitive demand. With an appropriate trial design this could both help elucidate the individual contribution of cognitive demand as well as methodologies with which to implement it in the clinic.

In conclusion, existing etiological models of ARCD and ARDs fail to adequately explain the underlying disease processes in a manner that (i) encompasses all the critical clinical observations of disease heterogeneity or trajectory, or (ii) has resulted in targeted interventions that meaningfully improve patient outcomes. In contrast, consideration of the environmental and evolutionary forces of cognitive demand leads to a holistic and multidimensional demand-driven model of ARCD and ARD etiology with both epidemiological and mechanistic support. This theoretical model allows for direct, simple, and equitable intervention points, as well as the possibility of improved outcomes in future pharmaceutical trials via inclusion of a holistic strategy that incorporates a component of cognitive demand to drive neurological homeostasis and repair. It also highlights the importance of a lifelong program of ongoing cognitive challenge coupled with diet and lifestyle factors, such as nutrient adequacy and sleep [37,44,48,154], that support the tissue-preserving adaptations those challenges stimulate. As the burden of AD is projected to cripple healthcare systems worldwide in the coming decades, we believe that this critical reappraisal of our approach is essential to maximizing our odds of success in preventing and treating the most prevalent forms of age-related cognitive decline and dementia.

## Figures and Tables

**Figure 1 cells-11-02789-f001:**
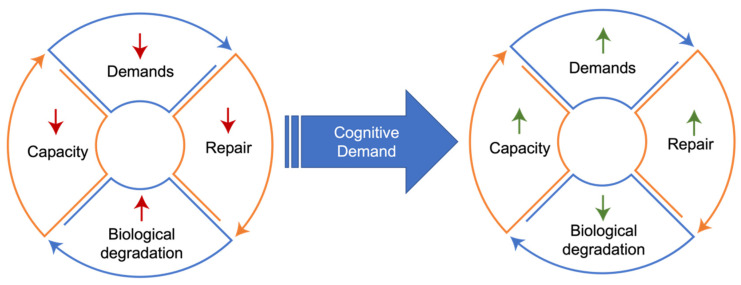
**Demand-function coupling.** Lack of cognitive demand drives a reduction in coupled reparative processes, ultimately resulting in less function and less capacity to initiate demand in a feed-forward process. Increased cognitive demand can reverse these processes, improving homeostasis and function.

**Figure 2 cells-11-02789-f002:**
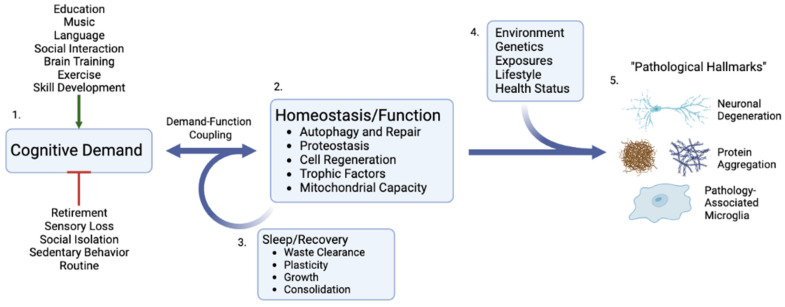
**Cognitive demand as the critical upstream determinant of cognitive function and decline.** 1. Cognitive demand is increased or decreased as a result of numerous individual and societal factors that can either positively or negatively affect the entire cascade. 2. As a result of direct demand-function coupling, cellular and organ responses down/upregulate homeostatic processes that can either increase or decrease repair, autophagy, and function. 3. Critical demand–function responses such as waste clearance and plasticity occur during recovery periods, particularly sleep. 4. In the setting of low demand and/or inadequate recovery, an individual’s personal environmental, health, and genetic circumstances result in differing and heterogeneous pathological phenotypes and cellular hallmarks of dementia and cognitive decline. 5. Created with BioRender.com.

**Figure 3 cells-11-02789-f003:**
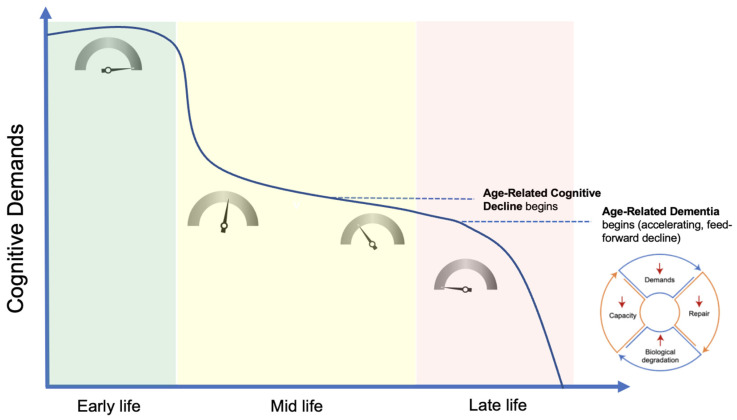
**Demand-driven model.** In early life, numerous factors, most notably the neuro-developmental program, maintain maximal cognitive demands and, via demand coupling, a maximal growth and repair signal (as indicated by the inset dials). Relative to early life, demands drop considerably in mid-life, leading to a steady deterioration in tissue structure and function, along with the onset of age-related cognitive decline. Once deterioration in tissue structure significantly constrains the scope of potential cognitive demands, a non-linear, accelerating rate of decline ensues, progressing to clinical dementia. Created with BioRender.com.

## Data Availability

Not applicable.

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
