# Peer review of "Demand Coupling Drives Neurodegeneration: A Model of Age-Related Cognitive Decline and Dementia"

_cells, 2022, doi:10.3390/cells11182789_

Round 1
Reviewer 1 Report
This Review, entitled “Demand-driven theory of age-related cognitive decline and dementia” proposes an interesting model which focuses on the contribution of cognitive stimulation and physical inputs that promote physical regeneration in neuroprotection from Age-related dementia (ARD) and cognitive decline, with emphasis on Alzheimer’s disease (AD). The Review proposes that downstream molecular events and pathways (such as increased ROS and inflammation with meningitis and sleep deprivation) manifest in differing effects at the cellular, tissue and behavioral level, and that effects from the top down, from behavior to environment can impact considerably down to the molecular and cellular pathology. The core of this model acknowledges that while risk factors (such as APOE) contribute to increased risk in ARD, physical (exercise) and mental (language, music, social and sensory inputs) stimulation are considerable protective factors. The authors cite societal differences in modernized and primitive culture on lifespan and ARD prevalence, as well as lifestyle and levels of activeness on ARD, which implicates the importance of top-down factors and their role in therapeutic ARD intervention.
The Review is very well written, and while the notion that lifestyle, physical activeness/exercise, education/cognitive stimulation and social inputs can impact on dementia risk is not entirely novel, the discussion and summary of various instances where top-down effects could impact on ARD pathology and pathogenesis is appreciated. A few comments should be considered below.
Comments:
A more concise, and descriptive title would be more informative; “demand-driven theory” is somewhat ambiguous. An alternative could be “Contribution of physical and mental stimulation on age-related cognitive decline and dementia”.
In regard to the above comment, “theory” may be an overclassification of the model presented.
Although there are many controversial aspects to the “amyloid hypothesis,” it must be acknowledged that the familial forms of early onset AD (fAD) are almost fully-penetrant, and cannot be compared directly to sporadic AD or ARD. Importantly, environmental and “top-down” effects are less likely to impact fAD, which due to the nature of the mutations in APP or PS1/PS2, are invariably associated with Abeta. Thus, the discussion here would be relevant to sporadic AD or dementia.
The authors may want to reconsider use of the word “demand”, and perhaps use descriptors such as “activity” or “stimulation”. At the very least, the word “demand” requires elaboration (cognitive, physical, or other).
There is no callout for Fig. 3. While it may be argued that cognitive “demand” is high during development, the drop in this demand during mid-life is variable between individuals and their lifestyle. Further, the sporadic nature of dementia (especially AD) needs to be integrated into the model, and many factors in addition to mental demands or stimulation contribute to dementia risk or onset.
A reference citation for the statement that the human brain is fully matured in individuals in their 30’s would be helpful (line 438).
Author Response
Authors’ response: We thank the reviewer for their positive and insightful comments, and hope that their concerns have been adequately addressed.
Comments:
A more concise, and descriptive title would be more informative; “demand-driven theory” is somewhat ambiguous. An alternative could be “Contribution of physical and mental stimulation on age-related cognitive decline and dementia”.
In regard to the above comment, “theory” may be an overclassification of the model presented.
Authors’ response: We agree that the title could be clearer and that use of the word “theory” overstates the current position. The title has been changed, and we have used the word “model” instead of “theory” throughout the revised manuscript.
Although there are many controversial aspects to the “amyloid hypothesis,” it must be acknowledged that the familial forms of early onset AD (fAD) are almost fully-penetrant, and cannot be compared directly to sporadic AD or ARD. Importantly, environmental and “top-down” effects are less likely to impact fAD, which due to the nature of the mutations in APP or PS1/PS2, are invariably associated with Abeta. Thus, the discussion here would be relevant to sporadic AD or dementia.
Authors’ response: We completely agree and apologise that this was not made clearer. The following sentence has been added to section 2: “Within this article we will therefore only consider a model for ARD, as true familial AD must be considered a separate entity.”
The authors may want to reconsider use of the word “demand”, and perhaps use descriptors such as “activity” or “stimulation”. At the very least, the word “demand” requires elaboration (cognitive, physical, or other).
Authors’ response: We feel the word demand is still appropriate, as has been used by others (e.g. Park et al. 2014. “The Impact of Sustained Engagement on Cognitive Function in Older Adults: The Synapse Project.” Psychological Science 25 (1): 103–12.). However, we have added a definition of demand to section 5: “To draw commonalities across physiologic systems, we will define demand as the application of a stimulus to a tissue, such as skeletal muscle or the brain, that requires adaptation in the form of structural or biochemical changes to increase function above current abilities.”
There is no callout for Fig. 3. While it may be argued that cognitive “demand” is high during development, the drop in this demand during mid-life is variable between individuals and their lifestyle. Further, the sporadic nature of dementia (especially AD) needs to be integrated into the model, and many factors in addition to mental demands or stimulation contribute to dementia risk or onset.
Authors’ response: We have added a callout to figure 3 and expanded on the relative importance of other factors throughout the revision, including how these can be assessed with modern clinical trial design. However, the nature of the model still rests on demand as the major upstream regulator of neurological function.
Reviewer 2 Report
In this very interesting review the authors propose the Demand-Driven Theory of Age-Related Cognitive Decline and Dementia.
The authors show the differences in the clinical course and risk factors between classical early-onset or familial AD and “late onset” AD.
The central element of this theory is the Demand-function coupling
I suggest the authors to deepen some elements of their interesting article
1. Why are early-onset dementia cases phenotypically similar to those with a late dementia?
2. What can be the clinical utility of this model? How is it possible to act in the ARCD phase for a reversal of the phenomenon?
3. In a EBM logic how the holistic therapeutic approach may be assessed ?
Minor point
In the introduction the term ARCD should be clarified
Author Response
Authors’ response: We appreciate the critical appraisal from the viewpoint of the background provided and fast-moving nature of the field. While expanding broadly on the various hypotheses that explain AD aetiology is beyond the scope of the review, we hope that the additional and updated references provide some robustness to the case being made.
The authors state in the introduction that there have been no significant advances in recent years in the treatment of AD, which is true, but I recommend that the authors use more up-to-date literature discussing the use of antibodies and other novel therapies.
The authors do not use a bibliography in the introduction (only two references) and make some statements that require bibliographic references. An introduction to an article (including reviews) should be very well argued and justified.
Authors’ response: The introduction has been reorganized to clarify where original claims are being made. It has also been lengthened slightly and multiple citations have been added, including recent reviews of the scope of future pharmacological therapies for AD.
The PSEN1 mutation in Auguste Deter's DNA is still a matter of debate as was shown later (https://alz-journals.onlinelibrary.wiley.com/doi/10.1016/j.jalz.2014.06.005).
Authors’ response: This sentence has been edited to add this reference, and now reads: “the first case described by Alois Alzheimer, Auguste Deter, was supposedly found to have a mutation in the presenilin (PSEN) 1 gene, though this has also been disputed.”
The authors said "This suggestion is underscored by the fact that plaque burden is poorly associated with "late onset" AD severity and progression, in both humans and animal models". I think that in the last 7 years many articles have been published that enrich the debate mentioned by the authors, so I recommend including updated bibliography if they want to give robustness to their approach.
Authors’ response: A second sentence with citations up to and including 2022 has been added here that reads “Though a thorough examination of the amyloid cascade hypothesis is beyond the scope of this review, it must be noted that multiple alternative models are emerging to attempt to explain the wide variety of plaque burden seen relative to an individual’s trajectory of cognitive function.”
In point number 3 I consider that the authors deviate from the central theme of the article, although the examples are interesting, I think they could be summarised in a remarkable way.
Authors’ response: We agree, and as a result have shortened this section and removed the table.
Throughout the text the authors make statements that require bibliographical references to support them. Examples: pos 160, 258, 274, 302, 319, 393, 400, Point 7.1, etc. In order for this article to be published, the authors should revise the text and include appropriate references to aid understanding of the text.
Authors’ response: Citations, often with some additional clarifying text, have been provided at each of these points.
Point 7.4. The review puts forward an interesting hypothesis, but I consider that it sometimes fails to take into account fundamental disease factors, including those related to the environment. An AD hypothesis should encompass everything we know about the disease. I recommend to the authors that when they make claims such as those raised in this point, they indicate that there are other relevant environmental, lifestyle, genetic, etc. factors.
Authors’ response: We agree with the reviewer that the environment is critical, and have incorporated it at multiple steps in the description as well as the summary figure (Figure 2). However, the intention of this section is to review the various factors that contribute to a reduction in cognitive stimulation over the lifespan rather than a review of all environmental factors that contribute. The role of established lifestyle factors is addressed in section 9, which summarizes the model. We have also added the following sentence to section 4: “And while our model below proposes that cognitive demand may be the critical upstream factor driving the reliance and influence of other environmental factors on cognitive outcomes, we must acknowledge the importance of overall lifestyle and the environment on the risk of ARD and ARCD.”
I agree that a holistic approach is needed, which should also include current therapeutic approaches, although in this respect the authors should make clear the difference between prevention and treatment of the disease. On the other hand, I think that the authors do not mention any of the FINGER studies that examine precisely these kinds of prevention approaches.
Authors’ response: This is an important point and an oversight on our part. The FINGER trial is now specifically mentioned in section 8, and is referenced at multiple times throughout the manuscript.
The authors should include a section where they discuss the limitations of the hypothesis put forward in this paper.
Authors’ response: A limitations an implementation paragraph has been added to section 10.
Pos 510 DDToARCD????
Pos 554 "cognitive challenge.(124-127)".
Authors’ response: These have been edited. The former was our own shorthand that was not removed during final editing.
Bibliography. Authors should reduce the number of authors appearing in the bibliographic citations.
Ref 65 is all that text necessary? what does it mean?
Authors’ response: The bibliographic output has been changed, which we hope has solved these issues.
Reviewer 3 Report
The authors put forward an interesting paper on a topical issue. I agree with the approach, and believe that new perspectives on Alzheimer's disease are needed. This type of work is important to generate a debate on the causes of this complex disease. I consider that there are parts of the text in which they overdevelop some concepts that are not relevant to the understanding of the review and that make the reading somewhat more complex. I recommend the authors to reduce those parts that are less related to the AD and the hypothesis they put forward. In addition, a review article must have good bibliographical support, the main problem I find in this review is the lack of references to support some claims made by the authors.
Below are some aspects that need to be revised by the authors in order for the article to be published in the Cells review.
The authors state in the introduction that there have been no significant advances in recent years in the treatment of AD, which is true, but I recommend that the authors use more up-to-date literature discussing the use of antibodies and other novel therapies.
The authors do not use a bibliography in the introduction (only two references) and make some statements that require bibliographic references. An introduction to an article (including reviews) should be very well argued and justified.
The PSEN1 mutation in Auguste Deter's DNA is still a matter of debate as was shown later (https://alz-journals.onlinelibrary.wiley.com/doi/10.1016/j.jalz.2014.06.005).
The authors said "This suggestion is underscored by the fact that plaque burden is poorly associated with "late onset" AD severity and progression, in both humans and animal models". I think that in the last 7 years many articles have been published that enrich the debate mentioned by the authors, so I recommend including updated bibliography if they want to give robustness to their approach.
In point number 3 I consider that the authors deviate from the central theme of the article, although the examples are interesting, I think they could be summarised in a remarkable way.
Throughout the text the authors make statements that require bibliographical references to support them. Examples: pos 160, 258, 274, 302, 319, 393, 400, Point 7.1, etc. In order for this article to be published, the authors should revise the text and include appropriate references to aid understanding of the text.
Point 7.4. The review puts forward an interesting hypothesis, but I consider that it sometimes fails to take into account fundamental disease factors, including those related to the environment. An AD hypothesis should encompass everything we know about the disease. I recommend to the authors that when they make claims such as those raised in this point, they indicate that there are other relevant environmental, lifestyle, genetic, etc. factors.
I agree that a holistic approach is needed, which should also include current therapeutic approaches, although in this respect the authors should make clear the difference between prevention and treatment of the disease. On the other hand, I think that the authors do not mention any of the FINGER studies that examine precisely these kinds of prevention approaches.
The authors should include a section where they discuss the limitations of the hypothesis put forward in this paper.
Pos 510 DDToARCD????
Pos 554 "cognitive challenge.(124-127)".
Bibliography. Authors should reduce the number of authors appearing in the bibliographic citations.
Ref 65 is all that text necessary? what does it mean?
Author Response
Authors’ response: We thank the reviewer for their insightful questions, answers to which have been added during revision.
- Why are early-onset dementia cases phenotypically similar to those with a late dementia
Authors’ response: A sentence has been added to section 2: “). Similarly, once they reach their late stages, early- and late-onset AD share multiple phenotypic commonalties, presumably due to a final convergence of cerebral atrophy and loss of functional capacity that results in similar effects on higher cognitive functions.”
- What can be the clinical utility of this model? How is it possible to act in the ARCD phase for a reversal of the phenomenon?
- In a EBM logic how the holistic therapeutic approach may be assessed
Authors’ response: A new paragraph has been added to section 10 that we hope begins to cover both of these questions within the scope of the manuscript. It reads: “Though we believe that the demand model is holistic and explains the vast majority of observations relevant to ARCD and ARD, we must acknowledge some limitations and potential hurdles to implementation. In its current form as presented, cognitive demand is difficult to quantify, and therefore the model is difficult to test. From an implementation standpoint, one might hope that better public awareness of the importance of cognitive demand could drive behavior change and increased cognitive demand across the lifespan at the individual level; however, similar approaches to physical activity have yet to be broadly successful and personalized interventions are likely to be needed. Overall, it is also not yet known whether cognitive demand is both necessary and sufficient to explain ARCD, and any current attempt to prevent or reverse cognitive decline in a holistic manner should include multi-modal interventions based on available best evidence such as that from the FINGER trial. This hampers the ability to dissect the relevant contributions of different lifestyle factors, including cognitive demand, on the risk of ARCD and ARD. To develop the idea of cognitive demand into an evidence-based medicine framework, one potential idea would be to incorporate it into more modern adaptive trial designs such as SMARTs (Sequential Multiple Assignment Randomized Trials), which are increasingly being implemented in the field of lifestyle and behavior change research. An individual’s risk in multiple areas of lifestyle could assessed, and then participants randomized at multiple timepoints to sequential new interventions including personalized interventions to increase cognitive demand. With an appropriate trial design this could both help elucidate the individual contribution of cognitive demand as well as methodologies with which to implement it in the clinic.”
Minor point
In the introduction the term ARCD should be clarified
Authors’ response: This has been done, thank you.
Round 2
Reviewer 3 Report
I appreciate the authors' thorough revision of their work. I think it is now much better understood. The inclusion of references in the text makes it much easier to follow the work and to find previous sources. An article of this type requires good bibliographical support.
I consider that the authors have fulfilled my initial requirements. This kind of work is important to change our perspective on Alzheimer's disease and similar pathologies. I encourage the authors to continue working on the model proposed in this article. There is currently a lot of effort being put into understanding how lifestyle changes can prevent the development of the disease or at least delay it, so this model may receive a lot of input from such studies in the coming years.